# Relevant Aspects of the Dental Implant Design on the Insertion Torque, Resonance Frequency Analysis (RFA) and Micromobility: An In Vitro Study

**DOI:** 10.3390/jcm12030855

**Published:** 2023-01-20

**Authors:** Mariano Herrero-Climent, Artur Falcao, Joao Tondela, Aritza Brizuela, Blanca Rios-Carrasco, Javier Gil

**Affiliations:** 1Porto Dental Institute, Av. de Montevideu 810, 4150-518 Porto, Portugal; 2Centre for Innovation and Research in Oral Sciences (CIROS), Faculty of Medicine, University of Coimbra, Rua Larga 2, 3004-531 Coimbra, Portugal; 3Facultad de Odontología, Universidad Europea Miguel de Cervantes, C/del Padre Julio Chevalier 2, 47012 Valladolid, Spain; 4Department of Periodontology, Faculty of Dentsitry, University of Seville, 41009 Sevilla, Spain; 5Bioengineering Institute of Technology, Facultad de Medicina y Ciencias de la Salud, International University of Catalonia, Sant Cugat del Vallés, 08195 Barcelona, Spain

**Keywords:** dental implant, torque insertion, external connection, internal connection, micromobility, resonance frequency analysis

## Abstract

The major problems for the osseointegration of dental implants are the loosening of the screw that fixes the dental implant to the abutment and the micromovements that are generated when mechanical loads are applied. In this work, torque differences in the tightening and loosening of the connection screws after 1 cycle, 10 cycles and 1000 cycles for 4 dental implants with 2 external and 2 internal connections were analyzed. The loosening of 240 implants (60 for each system) was determined using high-precision torsimeters and an electromechanical testing machine. A total of 60 dental implants for each of the 4 systems were inserted into fresh bovine bone to determine the micromovements. The implant stability values (ISQ) were determined by RFA. The mechanical loads were performed at 30° from 20 N to 200 N. By means of the Q-star technique, the micromovements were determined. It was observed that, for a few cycles, the loosening of the screw did not exceed a loss of tightening of 10% for both connections. However, for 1000 cycles, the loss for the external connection was around 20% and for the internal connection it was 13%. The micromovements showed a lineal increase with the applied load for the implant systems studied. An external connection presented greater micromotions for each level of applied load and lower ISQ values than internal ones. An excellent lineal correlation between the ISQ and micromobility was observed. These results may be very useful for clinicians in the selection of the type of dental implant, depending on the masticatory load of the patient as well as the consequences of the insertion torque of the dental implant and its revisions.

## 1. Introduction

Dental implant healing during the initial osteointegration stage requires mechanical stability [1]; that is, it will only tolerate micromotions that do not exceed a 50 to 150 µm range or a fibrous interface may form around the implant, thus resulting in implant failure [2]. This primary stability, resulting from the mechanical engagement of the implant in the prepared surrounding bone [3], is often associated with implant insertion torque. However, this insertion torque cannot be excessive and each implant manufacturer recommends a certain maximum ranging from 30 to 70 N.cm [1]. On the other hand, insertion torques as low as 25 N.cm have been shown to be sufficient even for immediate loading protocols [4]. An implant stability assessment is commonly measured by a resonance frequency analysis (RFA), a method that measures the oscillation frequency of the implant in the bone, converting the values from hertz to the implant stability quotient (ISQ) [1,5,6,7,8,9]. Several parameters can influence the obtained insertion torque and ISQ values such as the implant design, bone quality and surgical bone preparation technique employed [10,11]. In this context, the relationship between these variables still requires an extended clarification.

Numerous studies have determined the relationship between dental implant designs and implant stability [12,13,14,15,16,17,18,19,20]. It was ascertained that different macro-designs of dental implants affect the stability values. However, especially important is the roughness; this improves the ISQ values in an important way because there is much more bone that is anchored to the implant, causing greater stability. A rough topography also favors secondary stability [21,22,23,24,25]. Another factor is the quality of the bone; it is possible to determine how a lower bone density causes less bone stability. The cortical bone strongly improves this stability [26,27]. Falco et al. demonstrated that large-thread implant designs are highly desirable in cases of poor bone quality [28]. Each implant geometry generates an insertion torque value, which is correlated to the stability of that specific implant in a specific bone quality, but the insertion torque is not an objective value to compare the primary stability between different implant types.

One of the most serious problems associated with the restorative aspect of dental implants is the loosening and fracturing of screws. Winkler et al. [29] advised that implant screws should be retightened 10 min after the initial torque application as a routine clinical procedure to help compensate for the settling effect. Mechanical torque gauges should be used instead of hand drivers to ensure the consistent tightening of the implant components to the torque values recommended by the implant manufacturers. In addition, Siamos et al. [30] proposed that an increase in the torque value for abutment screws above 30 N.cm could be beneficial for abutment implant stability and could decrease screw loosening episodes. Several authors have studied insertion torques [31,32] with other prosthetic materials. No significant differences were observed in the loading protocols of dental implants, although care should be taken with immediately loaded implants because the compressive stress when anchoring the implants to the cortical bone is occasionally so great that it causes a loss of vascularization of the bone [30,31]. No influence has been observed in bone volume augmentation techniques [32,33,34,35,36,37].

In this work, we studied two important aspects for stability: insertion torques and torque loss due to tightening and untightening processes, and stability as a function of the successive application of mechanical loads on dental implants with conventional designs with two types of connection. The results may help clinicians to know the influence of the masticatory load and the implant abutment connection system for the selection of the best implant system for each patient.

## 2. Materials and Methods

### 2.1. Dental Implants

Four types of dental implant connections using Klockner^®^ dental implants (SOADCO, Escaldes Engordany, Andorra) were used for the study. These implants were manufactured from commercial grade 3 titanium.

Two systems of connection for the implants with an abutment were evaluated, as shown in Figure 1:-External connection: SK2 implant (S) and KL implant (K).-Internal connection: VEGA^®^ implant (V) and ESSENTIAL^®^ implant (E).

The implants used were 4 mm in diameter and 12 mm in length.

The SK2 implant (S) has a machined collar that allows the connection gap to be elevated with respect to the bone crest. The conicity part offers a great primary stability, ending in a maximum diameter of 4.2 mm of the shoulder of the implant platform. The implants have a hexagonal connection that is 1.8 mm in height, with 3 mm between the planes.

KL implant (K) is a threaded, external connection implant with a slightly ogival shape and two sections at the tip to facilitate the surgical insertion. The implant presents as a connection a hexagon of 0.7 mm in height that allows the connection gap to be elevated with respect to the osseous crest. The hexagonal connection makes it possible to block the rotation and repositioning of the attachment. By means of the fixation screw, its complete immobilization is achieved. They end at a maximum diameter of 4.1 mm of the shoulder of the implant platform.

The ESSENTIAL^®^ cone implant (E) system has an internal double-loop connection. Its design at a cervical level is expected for placements following the semi-submerged technique, generating an optimal biological seal that prevents bone resorption caused by bacterial filtration through the connection gap.

The VEGA^®^ implant (V) is an internal connection implant that requires working with the implants at a bone level. The hexagonal polygon at the bottom of the cone facilitates the clinical handling and the correct positioning of the prosthetic components, minimizing rotational movements between the implant and prosthetic components.

### 2.2. Torque Tests

Sixty implants of each type were used. These were placed in a metallic support with the most coronal portion 2 mm outside the device, as shown in Figure 2. Fifteen dental implants of each type were used for each type of loading. Therefore, a total of one hundred eighty dental implants were used.

The abutment screws were made of Ti6Al4V. They were the same for all dental implants and the same screw was always used for each implant. As a consequence, one hundred eighty screws were used. No screw breakage occurred in the various tests due to the high mechanical strength of the titanium alloy used.

The torque loss of the connection screw was determined in three cases:A single tightening torque was applied to the screw to connect the abutment and implant at 30 N.cm. It was untightened by calculating the new torque.Multiple loads with the same tightening torque were applied. The tightening and untightening operation was performed 10 times, leaving 15 s between tightening and untightening.Cyclic loading, in which 1000 screw tightening and untightening operations were performed.

For the connection of the abutment to the implant, a cordless prosthetic screwdriver with a torque calibration system for screw fixation for NSK^®^ brand prosthetic procedures was used. The model used was iSD900^®^ (Tokyo, Japan). The measurements of the torque forces for screwing and unscrewing were determined with a high-precision Centor Touch Star TH^®^ torque tester (Andilog Technologies, Vitrolles, France). The sensitivity of the equipment was 0.015 N.m.

The cyclic load tests were performed using a Bionix servo-hydraulic testing machine, which was programmed to perform 1000 load cycles of 50 kg with an actuator speed of about 0.16 mm/s. The test frequency was 1.25 Hz. Figure 3 shows the performance of the cyclic load test.

### 2.3. Bone Quality

Fresh bovine rib was used, and the dental implants were inserted according to the indications of the commercial company. The bone type was determined by a scanning electron microscopy observation of the cortical and cancellous bone tissue zone obtaining class 2. The confirmation of this classification was performed by means of a Matzsusawa microhardness tester (Tokyo, Japan), which realized 15 microindentations in the cortical place of the bovine bone by applying 500 gf for 15 s with a Vickers indenter. The values obtained were 205 ± 19 MPa, which was classified as type 2 bone density [38].

### 2.4. Implant Stability Coefficient and Micromobility

Once the change in the insertion torques with the cycles were studied, the study of micromovements was carried out by applying a gradual tension to the dental implant. For this purpose, 60 implants of each of the 4 dental implant systems (2 with an external connection (S and K) and two with an internal connection (E and V)) were used. Incremental loads were applied from 20 to 200 N at an approximate angle of 30° direct to the implant-screwed provisional abutments (Figure 4).

The insertion torque was measured with an analog-calibrated dynamometer (BTG90CN-Tohnichi, Tokyo, Japan). The RFA was measured using a Penguin RFA (Integration Diagnostics Sweden AB, Göteborg, Sweden). This is a device used to measure implant stability by means of a resonance frequency analysis (RFA). A small magnetic measuring pencil, a MulTipeg^TM^, was used, which was screwed onto the implant or attachment and vibrated without contact. The measured value was represented as the implant stability coefficient (ISQ) and provided information about the appropriate restoration for the implant.

An electromechanical testing machine was used to apply the loads (Autograph AG-1 5 Kn Shimadzu, Tokyo, Japan). The images were taken at each 20 N load step by means of high-resolution Q-star systems, which had a sensitivity of 1 µm in the displacements. The quantification of the resulting micromotion was carried out by a high-resolution digital image correlation (DIC–VIC 3D Correlated Solutions). Figure 5 shows the test configuration.

### 2.5. Statistical Analysis

The data were reported by using the means, standard deviations (SDs), ranges, 95% confidence intervals (CIs) and medians (SPSS, SPSS Inc., Chicago, IL, USA). A paired two-sample *t*-test was performed. The results were considered to be significant at *p* < 0.001.

## 3. Results

Figure 6 shows the differences in the initial torque when performing a loading and unloading cycle. Figure 7 shows one corresponding with multiple cycling and Figure 8 shows one corresponding with cyclic cycling, which corresponded with 1000 cycles.

Once the dental implants were placed with their abutments, the RFA measurements were taken, obtaining mean ISQ values of 66 and 68 for the K and S dental implants, respectively. No statistically significant differences were observed in the values between the externally connected dental implants. However, the RFA values for the dental implants with an internal connection were 76 and 74 for the E and V implants, respectively. Among the dental implants with an internal connection, there were no statistically significant differences using, as in the previous case, values of *p* < 0.001. Statistically significant differences were observed between the two types of internal and external connection, with significantly higher values for the internal connection.

The values of the micromotion for each of the dental implants were obtained when a progressive load of 20 N was applied. It was observed that in all the systems, it increased in a linear way with the applied load and the micromotions were smaller when the ISQ was higher. The results for the K implants are shown in Figure 9, for the S implants in Figure 10, for the E implants in Figure 11, the results for the V bone level implants are shown in Figure 12.

The equations that related the micromotions to the applied force fitted the lines with correlation coefficients greater than 0.9 for each type of dental implant, as can be seen in Table 1.

## 4. Discussion

The first two types of insertion torque tests are common in clinics and, therefore, the results have an important clinical significance. In the case of 1000 cycles, it simulated the process of cyclic loads to which the connection screw between the implant and the abutment is subjected, which occasionally generates the loosening of the screw and even the fracture of the same. In the latter case, it simulated the fatigue processes to which the screw is subjected. From the results of the differences in the tightening and untightening torque of the dental implants, in the single and multiple cycling, the torque difference values between the four dental implant systems did not show statistically significant differences, with a value of *p* < 0.001. However, when we studied the cyclic behavior, with the implants subjected to 1000 cycles of tightening and loosening, the dental implants with an external connection had higher torque difference values to those corresponding with an internal connection. These differences were statistically significant, with a value of *p* < 0.001. There were also statistically significant differences between the V implants at the bone crest level and the E implants. This may have been because the V implants had a more severe sandblasting treatment on the entire dental implant, including the implant neck, which presented a greater surface energy of a compressive nature. This increase in the rough surface would produce an increase in fixation due to the frictional forces generated by the bone on the implant, making its movement more difficult. This fact has been studied by different authors [39,40], who determined that the level of the compressive state on the surface could exceed compressive residual stresses of more than 100 N, causing a closing stress that could justify these differences in the dental implant at the osseous level [40,41,42].

We did not observe settlement or friction. The screws made of Ti6Al4V had a greater hardness; thus, when unscrewed, they came out easily. Large deformations occur in commercially pure titanium and gold screws, which a few companies use to increase deformation and grip [43,44].

The values of the micromovement increased as the mechanical load increased in the studies carried out up to the application of a load of 200 N at an angle of 30°. As is well-known, mechanical loads cause small deformations in the implant and the implant transfers the load to the surrounding tissue, causing micromovements. It can be seen from the graphs in Figure 9, Figure 10, Figure 11 and Figure 12 that, in all cases, the micromotion produced linearly depended on the applied load. This linear relationship occurred for all types of implants, whether they had internal or external connections, with a correlation coefficient greater than 0.9 (Figure 13).

The slope of the straight line was higher in the straight lines of the externally connected dental implants, indicating that these dental implants had a higher sensitivity to the load application. For the same load, the external connection implants presented a greater amount of micromovement and this grew faster. These aspects should be considered by clinicians for the type of patients in whom the dental implants should be placed. In principle, for patients with bruxism or with a high application of mechanical stresses to the dental implant, it is better to place dental implants with an internal connection than an external connection.

A variable that was clearly affected was that as the ISQ values increased, the micromovements became smaller for the load levels studied. The slope of the straight line between the micromovement and applied force indicated the ease of the provoking micromovement when we applied a force. If we represented these slopes with respect to the ISQ values (Figure 13), we obtained a linear decreasing relationship. The higher the ISQ, the more the micromotion was impeded. With these equations, we could determine the micromotion that could occur for a given level of ISQ and a given type of implant for a given load level. The clinician’s knowledge of these relationships will be useful for the clinical evaluation and the convenience of placing a type of implant.

The results of our research agreed with those obtained by Bergamo et al. [11], who observed a relationship between the insertion torque and ISQ. Osseodensification improves the implant stability, irrespective of the arch and area operated on as well as the implant design and dimension, with exceptions for short implants and the connection type.

The study by Farronato et al. [26] about the influence of the RFA and applied loads complement our studies because they determined that the implant diameter was not associated with the RFA or insertion torque. These results suggest that the implant can achieve a good level of primary stability in terms of the insertion torque and the RFA. In addition, a strong correlation was found between the values of the insertion torque and the RFA. From the results of this work, we confirmed that this relationship was lineal. Similarly, Falco et al. [28] demonstrated that an implant abutment connection plays an important role in the degree of primary implant stability with the bone quality. Each implant geometry generated an insertion torque value, which was correlated to the stability of that specific implant in a specific bone quality. They explained that the insertion torque was not an objective value to compare the primary stability between different implant types. The results of Stacchi et al. [10] are also interesting; they confirmed that no significant differences in either the primary or secondary stability or the implant survival rate after 1 year of loading were demonstrated between implants inserted into sites prepared with osseodensification drills and a piezoelectric implant site preparation.

A limitation of this work was that we only performed the tests with a class 2 bone type and with a dental implant system that presented a certain design for internal and external connections. However, the different dental implants had similar designs and, in principle, should not have varied significantly in micromovement with respect to the applied loads. Another limitation of this study was that the dental implants were made of commercially pure titanium and, although these are the most common in the market, there are also other implants made of Ti6Al4V, Ti 13Zr or zirconia that may vary the results of this work [45,46].

In addition, the implant design must be taken into consideration when placed in a lower density bone and an insertion torque applied. It is very important to follow the recommendations of the manufacturer. The insertion torque must be considered as an important factor and a loss of torque with possible tightening and untightening cycles must be considered. A high torque value is important to achieve a good fixation and this load can be transferred to the tissue. However, excessive torque can cause plastic deformations in dental implants, especially in grade 1 and 2 titanium implants, which have a lower elastic limit. Cases of fractured dental implants have even been observed due to an excessive mechanical load applied in the torque.

With this study, we aimed to contribute to a better understanding of the variables for the fixation of dental implants and to improve their stability. There are many limitations to the study such as the difference in implant designs and the type of screws and in vivo tests could not be performed. Therefore, it is important to continue working on the variables that affect stability to improve the manufacture of dental implants and to provide clinicians with criteria to improve the long-term behavior of dental implants.

## 5. Conclusions

A decrease in the tightening of an abutment implant connection at a few cycles was observed. No differences were observed between external and internal connection implants. When 1000 cycles were reached, the decrease was around 20% with a greater loss for the external connection. The micromovements of the dental implant presented a linear relationship with the level of applied load, with a higher slope for the external connection than the internal ones. The ISQ values were higher for the internally connected implants and a linear relationship between the ISQ values and the increase in micromotion with the application of a mechanical load was demonstrated.

## Figures and Tables

**Figure 1 jcm-12-00855-f001:**
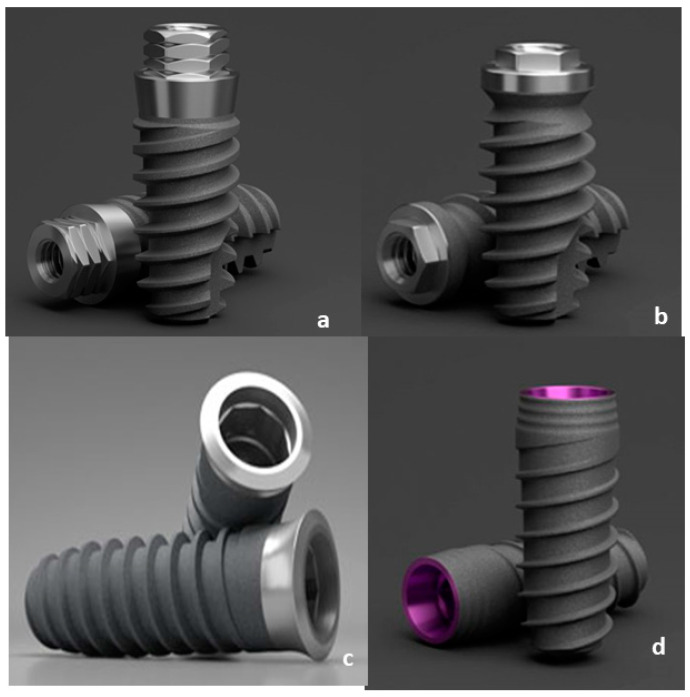
Implants studied: (**a**) SK2 dental implant (S); (**b**) KL dental implant (K); (**c**) ESSENTIAL^®^ dental implant (E); (**d**) VEGA^®^ dental implant (V).

**Figure 2 jcm-12-00855-f002:**
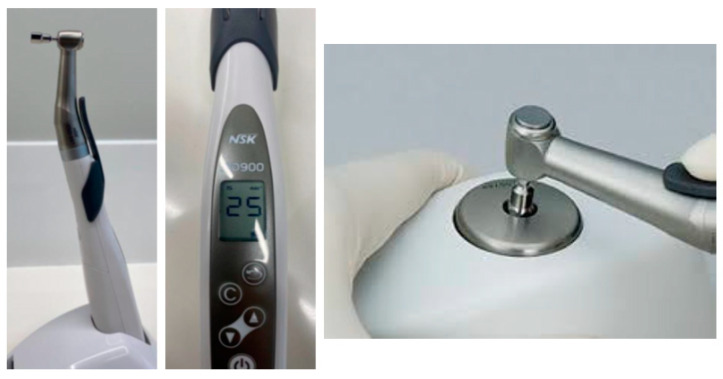
Prosthetic screwdriver with torque calibration system for screw fixation.

**Figure 3 jcm-12-00855-f003:**
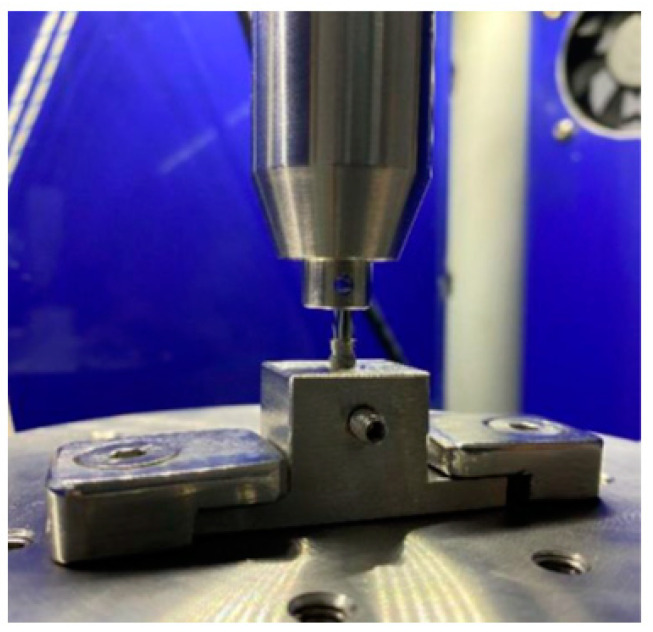
Fixation of the implant to cyclic load tests.

**Figure 4 jcm-12-00855-f004:**
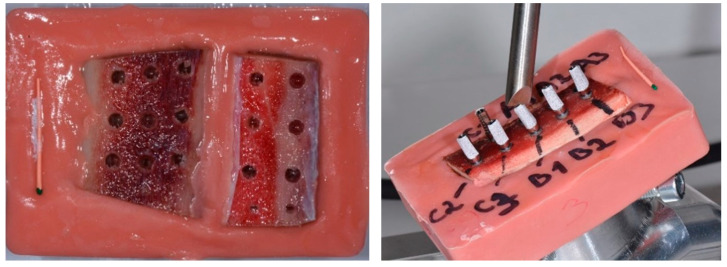
Fresh bovine bone (class 2) and loading process for each dental implant systems.

**Figure 5 jcm-12-00855-f005:**
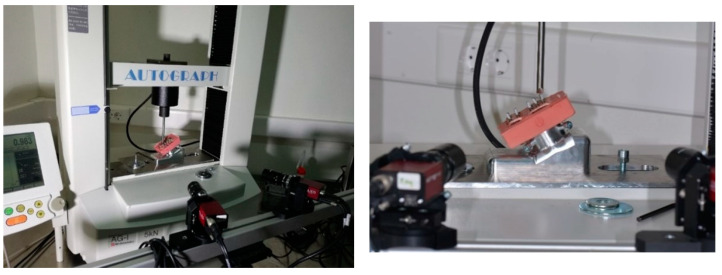
Configuration of the experiment in an electromechanical test machine that applied loads from 20 N to 200 N and the two high-resolution cameras that determined the micromovements.

**Figure 6 jcm-12-00855-f006:**
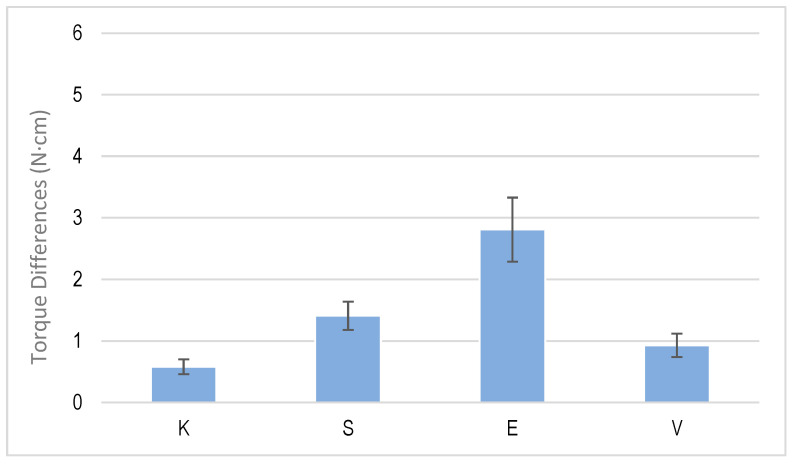
Torque differences for one cycle of tightening and untightening torque for each dental implant system.

**Figure 7 jcm-12-00855-f007:**
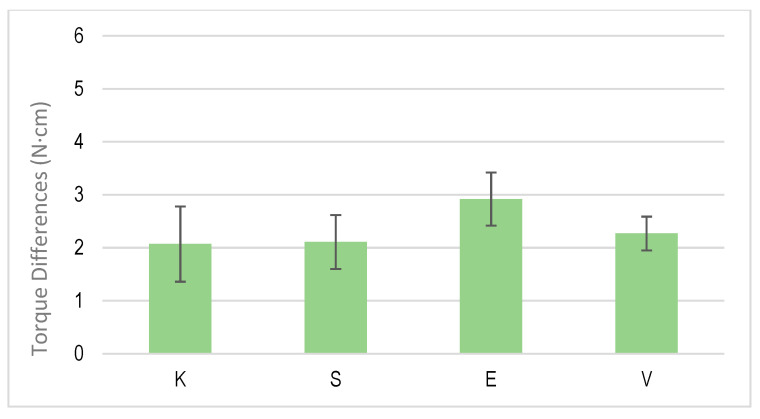
Torque differences for multiple cycles of tightening and untightening torque for each dental implant system.

**Figure 8 jcm-12-00855-f008:**
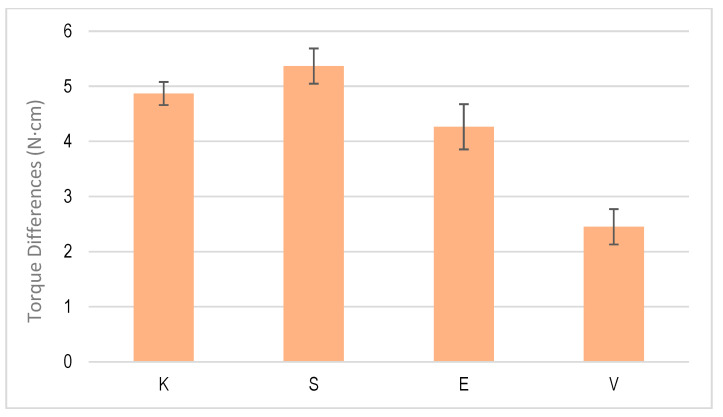
Torque differences for the cyclic test of tightening and untightening torque for each dental implant system.

**Figure 9 jcm-12-00855-f009:**
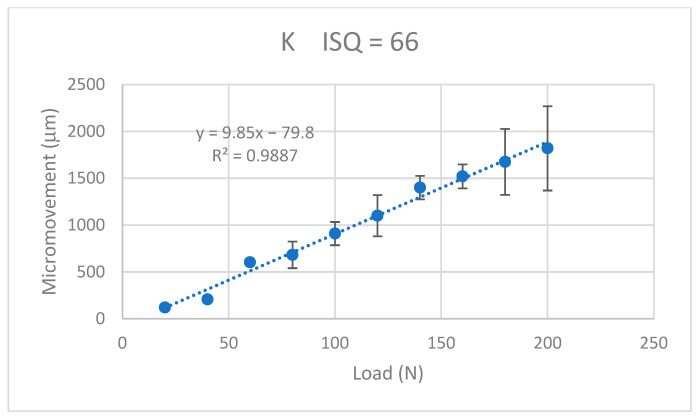
Micromovements in micrometers associated with the load in N for the K implants.

**Figure 10 jcm-12-00855-f010:**
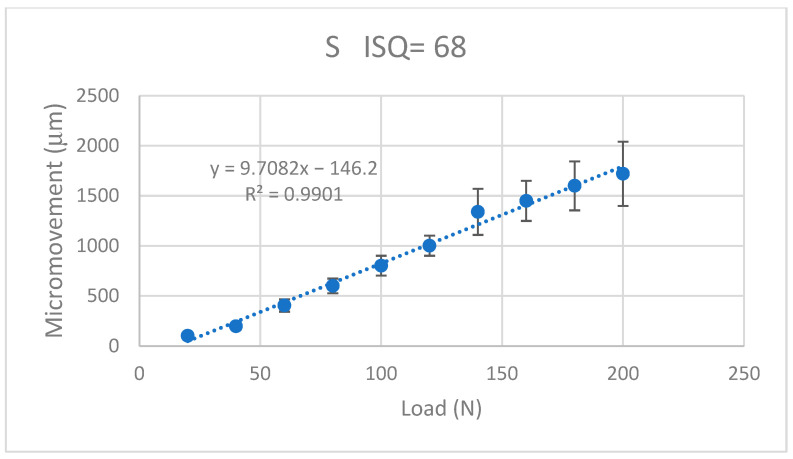
Micromovements in micrometers associated with the load in N for the S implants.

**Figure 11 jcm-12-00855-f011:**
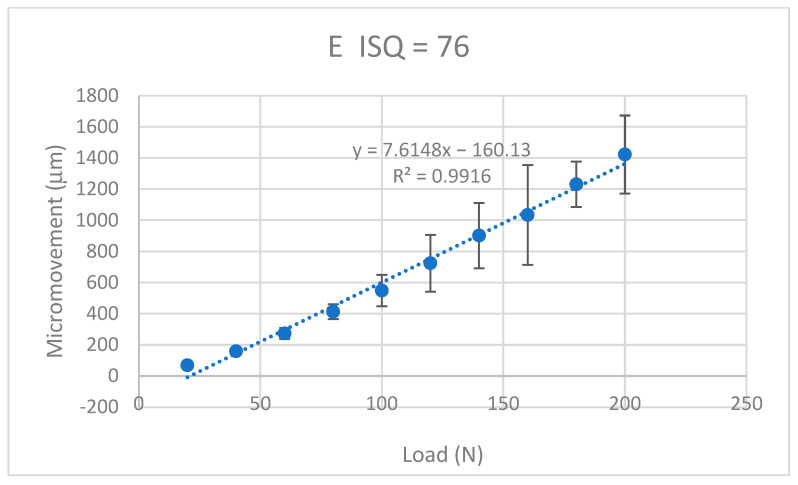
Micromovements in micrometers associated with the load in N for the E implants.

**Figure 12 jcm-12-00855-f012:**
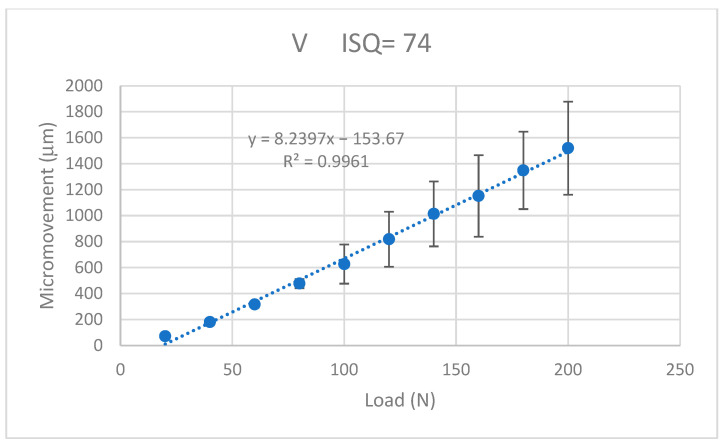
Micromovements in micrometers associated with the load in N for the V implants.

**Figure 13 jcm-12-00855-f013:**
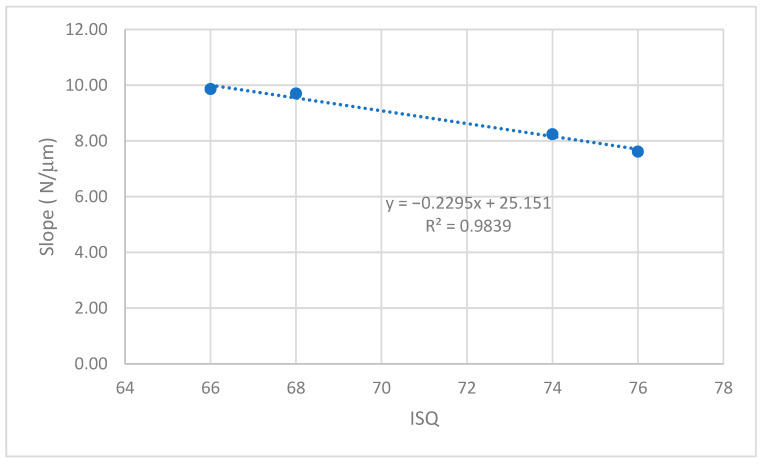
Relationship between the slope of the linear equation (micromovement-applied load) versus ISQ.

**Table 1 jcm-12-00855-t001:** Equations relating the micromovements (M) to the applied load (F) and their correlation coefficient.

Dental Implant	Equation	Correlation Coefficient
K	M = −79.8 + 9.850 F	0.9887
S	M = −146.2 + 9.708 F	0.9901
E	M = −160.1 + 7.615 F	0.9916
V	M = −153.7 + 8.239 F	0.9961

## Data Availability

The data are available upon request from the corresponding author.

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
