# Peer review of "Relevant Aspects of the Dental Implant Design on the Insertion Torque, Resonance Frequency Analysis (RFA) and Micromobility: An In Vitro Study"

_jcm, 2023, doi:10.3390/jcm12030855_

Round 1

Reviewer 1 Report

This a well planed study and the authors should be congratulated for it. The authors have a very extensive work and have not granted their research time/work, the manuscript it deserves. Their 240 implants with thousands of tightening and untightening cycles deserve more. I am sure they can do better and I hope my words can help

Line 73-76 One of the aspects that has not been studied is the influence of the external or internal connection of dental implants.

There are several authors that have studied insertion torques that I advise the authors to read and include in their manuscript:

Alikhasi M, Monzavi A, Bassir SH, Naini RB, Khosronedjad N, Keshavarz S. A comparison of precision of fit, rotational freedom, and torque loss with copy-milled zirconia and prefabricated titanium abutments. Int J Oral Maxillofac Implants. 2013;28(4):996–1002

 Siamos G, Winkler S, Boberick KG. Relationship between implant preload and screw loosening on implant-supported prostheses. J Oral Implantol. 2002 Jan;28(2):67–73

P. Bicudo, J. Reis, A.M. Deus, L. Reis, M.F. Vaz, Performance evaluation of dental implants: An experimental and numerical simulation study, Theoretical and Applied Fracture Mechanics, 2016, 85, Part A: 74-83,

Winkler, Sheldon et al. “Implant screw mechanics and the settling effect: overview.” The Journal of oral implantology vol. 29,5 (2003): 242-5. 

Line 155 reads  155 mN.m. please convert to n/cm as it's more common to use when talking about dental implants. Engineering wise it is correct N/m.

For the RFA:

Lee J, Pyo SW, Cho HJ, An JS, Lee JH, Koo KT, Lee YM. Comparison of implant stability measurements between a resonance frequency analysis device and a modified damping capacity analysis device: an in vitro study. J Periodontal Implant Sci. 2020 Feb 18;50(1):56-66. 

Pozzi A, Tallarico M, Moy PK. Three-year post-loading results of a randomised, controlled, split-mouth trial comparing implants with different prosthetic interfaces and design in partially posterior edentulous mandibles. Eur J Oral Implantol. 2014 Spring;7(1):47-61.

Wu AY, Huang HL, Hsu JT, Chee W. Biomechanical effects of the implant material and implant-abutment interface in immediately loaded small-diameter implants. Clin Oral Investig. 2014 May;18(4):1335-1341

The materials sections is very poorly explained and confusing. The authors are combining different experiments with similar names and procedures that are in fact very different. Also the authors have a not standard English nomenclature that doesn't help the reader. The article needs an English revision because some parts seem like a translation from their language. 

Unit load a tightening torque is applied to the abutment bolt at 30Ncm and it is untightened by calculating the new torque.

what does unit mean? What's an abutment?

Lines 85-135 the authors explain the implant geometry in length with little sense for the study and forget to explain the basic experimental unit that was employed

Please use abutment screw instead of bolt.

What was the rational for the tightening protocols? Are the authors suggesting clinicians tighten and loose the abutment screw 10 times?

How many screws were used? always the same one? with so many tightening and untightening cycles? no screw failed? this is unheard of.

Over time, screws tend to suffer from settling. Did the authors not observe this?

Lines 170-198 this is too confusing. First there's a load sequence, followed by insertion torque, then RFA measurements and load steps again.

Line 193 Translated with www.DeepL.com/Translator (free version) ????????

The authors write several times: No statistically significant differences but there are no statistical test or analysis described in Materials and Methods section. Please correct

Lines 260-272 are the authors trying to explain abutment screw (that connects to the inside of the implant ) problems with the sandblasting of the external part of the implant? I also think that the article is very confusing because the definitions used by the authors seem like interchangeable but aren't. Insertion torque, abutment screw torque, insertion stability, 

Lines 283-297 Are results and should be presented in the results section

Line 283 The equations fitted to the micromotion values in relation to the load applied every 283 20 N at 30º for each implant system...  makes little sense please clarify

Line 321 replace considerations of the commercial company with recommendations of the manufacturer  

Lines 322 -324 The insertion torque must be considered as an important factor and the loss of torque with possible tightening and untightening cycles must be taken into account. Where is the data that support this statement?

The discussion is non existant as the authors failed in identifying any previous studies. Please review the suggested literature.

The conclusions should be less results and more a take home message. The first few lines fail in doing so. I don't understand the last sentence based on the results

There's a thesis like sequence in this manuscript that needs to be solved. I advise the authors to remake the Materials and Methods and then the discussion.

Author Response

REVIEWER 1

Dear Reviewers,

Thanks for taking the time to review our manuscript and suggest to us to improve our work by providing a lot more detail. We have done so, and we are now submitting a manuscript that not only addresses the points you specifically raised but also many others that we have considered in order to deliver what we think is a much improved version of our work. This version includes more paragraphs, English grammar revisions in all main sections, new references. Thanks a lot and happy new year. We are looking forward to your comments.

Sincerely,

Francisco-Javier Gil Mur

This a well planed study and the authors should be congratulated for it. The authors have a very extensive work and have not granted their research time/work, the manuscript it deserves. Their 240 implants with thousands of tightening and untightening cycles deserve more. I am sure they can do better and I hope my words can help

  1. Line 73-76 One of the aspects that has not been studied is the influence of the external or internal connection of dental implants.

There are several authors that have studied insertion torques that I advise the authors to read and include in their manuscript:

Alikhasi M, Monzavi A, Bassir SH, Naini RB, Khosronedjad N, Keshavarz S. A comparison of precision of fit, rotational freedom, and torque loss with copy-milled zirconia and prefabricated titanium abutments. Int J Oral Maxillofac Implants. 2013;28(4):996–1002

Siamos G, Winkler S, Boberick KG. Relationship between implant preload and screw loosening on implant-supported prostheses. J Oral Implantol. 2002 Jan;28(2):67–73

  1. Bicudo, J. Reis, A.M. Deus, L. Reis, M.F. Vaz, Performance evaluation of dental implants: An experimental and numerical simulation study, Theoretical and Applied Fracture Mechanics, 2016, 85, Part A: 74-83,

Winkler, Sheldon et al. “Implant screw mechanics and the settling effect: overview.” The Journal of oral implantology vol. 29,5 (2003): 242-5.

The authors have added the references and have introduced a new paragraph with the most important results of these papers. Thank you for your help.

  1. Line 155 reads 155 mN.m. please convert to n/cm as it's more common to use when talking about dental implants. Engineering wise it is correct N/m.

Done

For the RFA:

Lee J, Pyo SW, Cho HJ, An JS, Lee JH, Koo KT, Lee YM. Comparison of implant stability measurements between a resonance frequency analysis device and a modified damping capacity analysis device: an in vitro study. J Periodontal Implant Sci. 2020 Feb 18;50(1):56-66.

Pozzi A, Tallarico M, Moy PK. Three-year post-loading results of a randomised, controlled, split-mouth trial comparing implants with different prosthetic interfaces and design in partially posterior edentulous mandibles. Eur J Oral Implantol. 2014 Spring;7(1):47-61.

Wu AY, Huang HL, Hsu JT, Chee W. Biomechanical effects of the implant material and implant-abutment interface in immediately loaded small-diameter implants. Clin Oral Investig. 2014 May;18(4):1335-1341

The references have been introduced in the text.

  1. The materials sections is very poorly explained and confusing. The authors are combining different experiments with similar names and procedures that are in fact very different. Also the authors have a not standard English nomenclature that doesn't help the reader. The article needs an English revision because some parts seem like a translation from their language.

The authors have improved the nomenclature, the text has been ordered, some paragraphs have been added for clarity and subdivisions have been made to improve comprehension.

  1. Unit load a tightening torque is applied to the abutment bolt at 30Ncm and it is untightened by calculating the new torque.

The sentence has been improved.

  1. what does unit mean? What's an abutment?

It is a mistake. A single tightening load. Sorry. Abutment is the prosthetic part. The connection is abutment-dental implant. The sentence has been corrected.

  1. Lines 85-135 the authors explain the implant geometry in length with little sense for the study and forget to explain the basic experimental unit that was employed

The text has been reduced by more than half, eliminating those features of dental implants that have no influence on the objectives of this manuscript.

  1. Please use abutment screw instead of bolt.

Done

  1. What was the rational for the tightening protocols? Are the authors suggesting clinicians tighten and loose the abutment screw 10 times?

The significance and clinical importance of this type of tests: single, multiple and cyclic has been discussed in the discussion of results in accordance with the reviewer's commentary.

  1. How many screws were used? always the same one? with so many tightening and untightening cycles? no screw failed? this is unheard of.

          These aspects have been introduced in Materials and Methods.

  1. Over time, screws tend to suffer from settling. Did the authors not observe this?

We have not observed settlement or friction, the screws being made of Ti6Al4V have greater hardness and when unscrewing they come out easily. Large deformations occur in commercially pure titanium and gold screws, which some companies use to increase deformation and grip. This explanation has been introduced in the text.

  1. Lines 170-198 this is too confusing. First there's a load sequence, followed by insertion torque, then RFA measurements and load steps again.

Yes. The text has been ordered. A new paragraph has been introduced to clarify and subdivisions has been incorporated to clarify Materials and Methods.

  1. Line 193 Translated with www.DeepL.com/Translator (free version) ????????

The sentence has been deleted.

  1. The authors write several times: No statistically significant differences but there are no statistical test or analysis described in Materials and Methods section. Please correct

Statistical tests have been introduced in the Materials and Methods.

  1. Lines 260-272 are the authors trying to explain abutment screw (that connects to the inside of the implant ) problems with the sandblasting of the external part of the implant? I also think that the article is very confusing because the definitions used by the authors seem like interchangeable but aren't. Insertion torque, abutment screw torque, insertion stability,

The text has been improved.

  1. Lines 283-297 Are results and should be presented in the results section

Done.

  1. Line 283 The equations fitted to the micromotion values in relation to the load applied every 283 20 N at 30º for each implant system... makes little sense please clarify

The sentence has been improved

  1. Line 321 replace considerations of the commercial company with recommendations of the manufacturer

Done

  1. Lines 322 -324 The insertion torque must be considered as an important factor and the loss of torque with possible tightening and untightening cycles must be taken into account. Where is the data that support this statement?

The discussion has been improved and justified according to the reviewer. Numerous papers have been referenced that justify and complement the work done in this manuscript.

  1. The discussion is non existent as the authors failed in identifying any previous studies. Please review the suggested literature.

The discussion has been improved and justified according to the reviewer. Numerous papers have been referenced that justify and complement the work done in this manuscript.

  1. The conclusions should be less results and more a take home message. The first few lines fail in doing so. I don't understand the last sentence based on the results

Conclusions have been rewritten according to the comment of the reviewer.

  1. There's a thesis like sequence in this manuscript that needs to be solved. I advise the authors to remake the Materials and Methods and then the discussion.

The authors have worked on the points that the reviewer has commented on. We are very grateful for his comments and suggestions and have tried to improve especially the Materials and methods and the discussion.

Reviewer 2 Report

Dear Authors,

The manuscript seems very interesting, detailed and offers much food for thought.

I have suggestions for improving the text and the scientific quality of the research.

English form: improve the grammar in some parts, I suggest having the text proofread by a native speaker.

Plagiarism: I suggest always checking for plagiarism with dedicated software

Abstract: the authors should structure the abstract better, dividing it into sessions and avoiding comments to be eventually inserted in the discussion. It also appears excessively long, delete a few sentences from the introduction.

Introduction: In the first sentences of the introduction I suggest referring to this important article on implant torque [doi: 10.11607/jomi.6285].

"One of the most used devices for ISQ monitoring the Penguin RFA (Integration Diagnostics 55 Sweden AB) have been introduced in the market." 

I suggest removing this sentence from the introduction and possibly discussing it in Materials and Methods.I suggest referring to recent studies on the importance of ISQ in vivo and the evaluation of this parameter [doi: 10.1111/cid.13140 - doi: 10.1111/cid.13007].

"Another factor is the quality of the bone, it was possible to de- 65 termine how the lower bone density causes less bone stability, being the cortical bone the 66 one that strongly improves this stability". Some references seem outdated and obsolete. I suggest referring to more recent studies on the influence of bone density on torque and ISQ [doi: 10.1563/aaid-joi-D-19-00145 - doi: 10.3390/dj8010021].

"One of the aspects that has not been studied is the influence of the external or internal 73 connection of dental implants." I do not agree with this sentence, I suggest deleting it.

"In this work we intend to study two important aspects for 74 stability: insertion torques and torque loss due to tightening and untightening processes 75 and stability as a function of the successive application of mechanical loads on dental im- 76 plants with conventional designs with two types of connection. For the determination of 77 stability, the RFA values were obtained by means of Penguin, one of the most sensitive 78 devices for this study, and the micromovements with high resolution Q-star systems that 79 allow to have a sensitivity of 1 micrometer in the displacements [8-10]. The results will 80 help the clinician to know the influence of the masticatory load and the implant-abutment 81 connection system for the selection of the best implant system for each patient." Rephrase this sentence. there must be no personal considerations in the introduction, moreover not validated by scientific references.

Considerations must be included in the discussion.

Methods: "Fresh bovine rib was used for them, and the dental implants were inserted according 174 to the indications of the commercial company. The bone type was class 2." how did you classify the bone density of the ribs? Have you used CT scans? Several studies have used different methods. The authors must specify this parameter.

"Translated with www.DeepL.com/Translator (free version)" line - 193

Check the entire text. The text must be proofread by a native speaker and do not use translation software.

The discussion part needs to be reviewed in detail. Comparisons with studies already in the literature should be included, discussing any differences or similarities in comparison with the results obtained in this study.

After careful review, the manuscript should be re-evaluated for possible publication.

Author Response

REVIEWER 2.

Dear Reviewer,

Thanks for taking the time to review our manuscript and suggest to us to improve our work by providing a lot more detail. We have done so, and we are now submitting a manuscript that not only addresses the points you specifically raised but also many others that we have considered in order to deliver what we think is a much improved version of our work. This version includes more paragraphs, English grammar revisions in all main sections, new references. Thanks a lot and happy new year. We are looking forward to your comments.

Sincerely,

Francisco-Javier Gil Mur

Dear Authors,

The manuscript seems very interesting, detailed and offers much food for thought.

I have suggestions for improving the text and the scientific quality of the research.

  1. English form: improve the grammar in some parts, I suggest having the text proofread by a native speaker.

            English has been revised

  1. Plagiarism: I suggest always checking for plagiarism with dedicated software.

I have revised the text by Turnitin with plagiarism lower than 5%.

  1. Abstract: the authors should structure the abstract better, dividing it into sessions and avoiding comments to be eventually inserted in the discussion. It also appears excessively long, delete a few sentences from the introduction.

According to the reviewer the abstract has been improved in its structure. In addition, the text has been reduced by 30%.

  1. Introduction: In the first sentences of the introduction I suggest referring to this important article on implant torque [doi: 10.11607/jomi.6285].

The authors have introduced the reference and a new paragraph about the results from the paper cited. Thank you

  1. "One of the most used devices for ISQ monitoring the Penguin RFA (Integration Diagnostics 55 Sweden AB) have been introduced in the market." I suggest removing this sentence from the introduction and possibly discussing it in Materials and Methods.I suggest referring to recent studies on the importance of ISQ in vivo and the evaluation of this parameter [doi: 10.1111/cid.13140 - doi: 10.1111/cid.13007].

In agreement with the reviewer, the text has been changed and the two references to ISQ suggested by the reviewer have been added. Thank you very much as it allows the reader to learn more about aspects of implant stability.

  1. "Another factor is the quality of the bone, it was possible to de- 65 termine how the lower bone density causes less bone stability, being the cortical bone the 66 one that strongly improves this stability". Some references seem outdated and obsolete. I suggest referring to more recent studies on the influence of bone density on torque and ISQ [doi: 10.1563/aaid-joi-D-19-00145 - doi: 10.3390/dj8010021].

The references have been introduced and the references obsolete have been deleted.

  1. "One of the aspects that has not been studied is the influence of the external or internal 73 connection of dental implants." I do not agree with this sentence, I suggest deleting it.

Done

  1. "In this work we intend to study two important aspects for 74 stability: insertion torques and torque loss due to tightening and untightening processes 75 and stability as a function of the successive application of mechanical loads on dental im- 76 plants with conventional designs with two types of connection. For the determination of 77 stability, the RFA values were obtained by means of Penguin, one of the most sensitive 78 devices for this study, and the micromovements with high resolution Q-star systems that 79 allow to have a sensitivity of 1 micrometer in the displacements [8-10]. The results will 80 help the clinician to know the influence of the masticatory load and the implant-abutment 81 connection system for the selection of the best implant system for each patient." Rephrase this sentence. there must be no personal considerations in the introduction, moreover not validated by scientific references.

Considerations must be included in the discussion.

The paragraph has been improved according to the reviewer

  1. Methods: "Fresh bovine rib was used for them, and the dental implants were inserted according 174 to the indications of the commercial company. The bone type was class 2." how did you classify the bone density of the ribs? Have you used CT scans? Several studies have used different methods. The authors must specify this parameter.

Class 2 was determined by scanning electron microscopy observation of the cortical and cancellous bone tissue zone and microhardness tests in cortical place obtaining values of 205 MPa, which is classified as type 2 bone density. This explanation with more detail has been introduced in Materials and Methods.

  1. "Translated with www.DeepL.com/Translator (free version)" line - 193.

The sentence has been deleted.

  1. Check the entire text. The text must be proofread by a native speaker and do not use translation software.

English has been revised

  1. The discussion part needs to be reviewed in detail. Comparisons with studies already in the literature should be included, discussing any differences or similarities in comparison with the results obtained in this study.

The discussion has been improved and justified according to the reviewer. Numerous papers have been referenced that justify and complement the work done in this manuscript.

After careful review, the manuscript should be re-evaluated for possible publication.

Round 2

Reviewer 2 Report

Authors improved manuscript quality and scientific soundness according to the reviewers' reccomandations